# Tiron Has Negative Effects on Osteogenic Differentiation via Mitochondrial Dysfunction in Human Periosteum-Derived Cells

**DOI:** 10.3390/ijms232214040

**Published:** 2022-11-14

**Authors:** Jin-Ho Park, Eun-Byeol Koh, Young-Jin Seo, Hye-Seong Oh, Ju-Yeong Won, Sun-Chul Hwang, June-Ho Byun

**Affiliations:** 1Department of Oral and Maxillofacial Surgery, Institute of Health Sciences, School of Medicine, Gyeongsang National University, Gyeongsang National University Hospital, Jinju 52727, Korea; 2Department of Convergence Medical Science, Gyeongsang National University, Jinju 52828, Korea; 3Department of Orthopaedic Surgery, Institute of Health Sciences, School of Medicine, Gyeongsang National University, Jinju 52828, Korea

**Keywords:** tiron, periosteum-derived cells, osteoblastic differentiation, mitochondria

## Abstract

Tiron is a potent antioxidant that counters the pathological effects of reactive oxygen species (ROS) production due to oxidative stress in various cell types. We examined the effects of tiron on mitochondrial function and osteoblastic differentiation in human periosteum-derived cells (hPDCs). Tiron increased mitochondrial activity and decreased senescence-associated β-galactosidase activity in hPDCs; however, it had a detrimental effect on osteoblastic differentiation by reducing alkaline phosphatase (ALP) activity and alizarin red-positive mineralization, regardless of H_2_O_2_ treatment. Osteoblast-differentiating hPDCs displayed increased ROS production compared with non-differentiating hPDCs, and treatment with tiron reduced ROS production in the differentiating cells. Antioxidants decreased the rates of oxygen consumption and ATP production, which are increased in hPDCs during osteoblastic differentiation. In addition, treatment with tiron reduced the levels of most mitochondrial proteins, which are increased in hPDCs during culture in osteogenic induction medium. These results suggest that tiron exerts negative effects on the osteoblastic differentiation of hPDCs by causing mitochondrial dysfunction.

## 1. Introduction

Oxidative stress caused by excessive reactive oxygen species (ROS) and limited antioxidant defense is involved in age-related conditions including cardiovascular disease, Alzheimer’s disease, cancer, and decreased regenerative potential of mesenchymal stem cells (MSCs) [1,2,3]. Hydroxyl and peroxyl radicals, hydrogen peroxide, and superoxide radical anions are generated in mitochondrial electron transport systems as byproducts of cellular metabolism [4]. ROS levels under physiological conditions are tightly regulated by various means, including the action of endogenous and exogenous antioxidants, resulting in a balance between ROS production and consumption [5]. In a normal environment, approximately 90% of the oxygen that enters a cell is used to produce energy via the mitochondrial respiratory chain [6,7,8,9,10]. Mitochondria play a pivotal role in the metabolism of trace elements, especially iron [11,12], and are major sources of ROS in most cell types; the constituents of the mitochondrial membrane are particularly susceptible to oxidative stress [4,13,14].

It is currently believed that basal levels of ROS are advantageous for stem cell proliferation and survival, whereas excessive levels of ROS induce oxidative stress in MSCs, leading to lipid and protein oxidation, DNA damage, premature aging, and cell death [15,16]. Stromal stem cells possess an innate characteristic of “stemness”, which decreases with age and with increasing passage numbers in cultured cells. As cultured cells undergo successive passages, ROS production and apoptosis gradually increase, while expression of antioxidant enzymes and capacity for differentiation gradually decrease [17,18]. The osteogenic capacity of senescent stem cells can be restored by treatment with antioxidants [19,20].

Antioxidants inhibit cellular senescence and improve the viability and differentiation of MSCs by reducing ROS generation. With regard to osteoblastogenesis, antioxidant deficiency-induced mitochondrial dysfunction may be the primary cause of osteoblast dysfunction [21,22]. The vitamin E analog tiron, or 4,5-dihydroxy-1,3-benzene disulfonic acid, is an antioxidant that penetrates the mitochondrial membrane and accumulates within mitochondria. The tiny size of tiron allows it to easily enter cells and modify intracellular electron transfer reactions by antioxidant mechanisms. Tiron is a potent antioxidant against the pathological effects of oxidative stress and can reverse ROS-induced cell injury. It is also a nontoxic chelator of a variety of metals, including iron, and is used to alleviate acute toxic metal overload in the body [23,24,25]. Tiron attenuated oxidative stress and inflammation in airways or acute pancreatitis models in a rat model, and the hepatoprotective effect of tiron against acetaminophen-induced acute liver injury has been reported. Moreover, previous studies have reported that the combination of tiron and tempol is also an effective treatment for pyrogallol- or hypoxia/reoxygenation-induced oxidative stress, and tiron reduced bortezomib-induced apoptosis in human lung cancer cells [26].

In bone metabolism, ROS-induced oxidative stress impairs skeletal integrity and reduces osteogenic differentiation of osteo-precursor cells and bone marrow-derived stromal cells. Although dissolved oxygen in blood has a limited diffusion distance (100–200 μm), it is effectively supplied to cells in native tissues via the fluid-rich extracellular matrix and very dense capillary networks. Physiological oxygen tension ranges from 1% to 13% in various tissues; however, conventional in vitro MSC culture is carried out in incubators equilibrated with 95% air and 5% CO_2_ (approximately 20% O_2_) [27,28]. Cultured MSCs reside in hypoxic niches where the low oxygen tension is thought to contribute to the maintenance of an undifferentiated state, but most transplanted MSCs experience oxidative stress and high ROS levels produced by host tissues or by the MSCs themselves [29]. Several studies suggest that ROS inhibit the osteogenic differentiation of MSCs [30,31]; however, other studies suggest that ROS and bone morphogenic proteins both enhance osteogenic MSC differentiation [32,33,34].

The effects of the ROS-scavenging antioxidant tiron on MSC differentiation into osteoblasts remain largely unknown. Therefore, we aimed to determine if tiron affects osteoblastogenesis in human periosteum-derived cells (hPDCs) with or without oxidative stress. We also examined the effects of tiron on mitochondrial function in hPDCs.

## 2. Results and Discussion

### 2.1. Treatment of hPDCs with Tiron

We treated hPDCs with various concentrations of tiron to determine the effects on cell viability. Treatment with 100 mM tiron clearly reduced the viability of hPDCs after 3 days of culture. Cells treated with 10 mM or greater concentrations of tiron had reduced viability compared with untreated cells after 7 days of culture; however, there was no significant difference in viability at any time point between untreated cells and cells treated with 1 mM tiron (Figure 1A). These results suggest that tiron at concentrations less than 1 mM does not affect the viability of hPDCs, but tiron at higher concentrations may reduce hPDC viability. Therefore, we treated cells with tiron at concentrations less than or equal to 1 mM in the remainder of our experiments.

Mitochondrial dysfunction plays a central role in the age-related deterioration of stem cell function and self-renewal in various tissues. Antioxidants protect mitochondria and cells from oxidative stress and damage. We measured the mitochondrial membrane potential (MMP) to assess changes in mitochondrial function in hPDCs treated with tiron. MMP is a key indicator of mitochondrial activity. A decline in MMP is associated with mitochondrial dysfunction that may lead to biological aging of cells and cell death. We found that hPDCs treated with 1 mM tiron had greater MMP than untreated hPDCs or hPDCs treated with 0.1 mM tiron, although the effect tended to decrease in a passage-dependent manner (Figure 1B).

Oxidative stress caused by hydrogen peroxide (H_2_O_2_) induces premature *senescence* within a short period of time *in many cell types.* To determine whether tiron has a protective function against H_2_O_2_-induced oxidative stress, we used β-galactosidase staining to evaluate cellular senescence in hPDCs treated with tiron. The hPDCs displayed irregular shapes and positive β-galactosidase staining, indicating senescence after exposure to H_2_O_2_. The senescence-associated β-galactosidase *activity* was clearly reduced when the cells were pretreated with 1 mM tiron before H_2_O_2_ exposure (Figure 1C).

Stem cell aging is a process in which stem cells progressively lose their ability for self-renewal or differentiation, undergo senescence, and eventually become functionally depleted. Mitochondrial dysfunction and oxidative stress are largely involved in aging and age-related diseases [35,36]. Damaged mitochondria produce excess ROS, leading to further oxidative stress, so enhancing mitochondrial function and reducing oxidative stress are logical strategies for restoring aged stem cell function [37]. Our results suggested that in hPDCs, the antioxidant properties of tiron increase mitochondrial activity and maintain cell functions by reducing senescence. Based on these results, we hypothesized that these protective effects of tiron could enhance the osteoblastic differentiation of hPDCs by slowing the stem cell aging process.

### 2.2. Effect of Tiron on Osteoblastic Differentiation of hPDCs

Control of ROS levels is important for regulating the fate of normal stem cells. Antioxidants prevent senescence and preserve the stemness of normal stem cells by reducing ROS generation during long-term culture [38,39]; however, evidence regarding the effects of tiron on MSC osteoblastic differentiation is limited. We examined the effects of tiron on the osteoblastic differentiation potential of hPDCs undergoing senescence due to serial passage in vitro.

Our previous study showed that hPDCs were positive for MSC markers and could differentiate in vitro into active osteoblastic cells capable of giving rise to the mineralization of a matrix [40,41]. In the present study, we found that the histochemical signal and activity of alkaline phosphatase (ALP), a marker of osteoblastic differentiation, in untreated hPDCs declined during serial passage in osteogenic-induction medium (Figure 2A,C). Similarly, alizarin red-positive mineralization of hPDCs declined in a passage-dependent manner in untreated hPDCs (Figure 2B,D). These results suggest that the osteoblastic differentiation capacity of hPDCs decreases during long-term culture, and in vitro passaging can be considered a primary reason for the decrease.

Treatment with tiron, regardless of the concentration, markedly reduced the ALP signal and activity in hPDCs undergoing senescence (Figure 2A,C). Similarly, alizarin red-positive mineralization was also reduced in cells treated with 0.1 mM or 1 mM tiron, albeit in a concentration-dependent manner (Figure 2B,D). The baseline osteoblastic activity of hPDCs cultured in osteogenic induction medium at passage 25 was very weak, so treatment with tiron had no additional effects on the osteogenic phenotype beyond that induced by the osteogenic induction medium (Figure 2).

Because antioxidants inhibit cellular senescence and enhance the in vitro differentiation of MSCs by reducing ROS generation, tiron might be expected to have favorable effects on the osteoblastic differentiation of hPDCs [42]. However, our results suggest that tiron has no direct effects on hPDC senescence and adversely affects hPDC osteoblastic differentiation.

### 2.3. Effect of Tiron on the Osteoblastic Phenotypes of hPDCs Treated with H_2_O_2_

We next examined the effects of tiron on the osteoblastic phenotypes of hPDCs under H_2_O_2_-induced oxidative stress. Increased oxidative stress and decreased capacity for antioxidant defense lead to decreased bone mineral density. Several studies showed that oxidative stress inhibits osteoblastic differentiation, increases osteoclast differentiation and function, and is a key factor underlying bone deficiency [43]. We found that exposure to H_2_O_2_ suppressed osteoblastic differentiation in hPDCs (Figure 3), which is in accordance with previous studies of MSC osteoblastic differentiation [39,40]. Treatment with 1 mM tiron, but not 0.1 mM tiron, further reduced ALP expression and activity in H_2_O_2_-treated hPDCs (Figure 3A,B). Similarly, pretreatment with H_2_O_2_ clearly reduced alizarin red-positive mineralization and calcium content in the hPDCs; however, treatment with 0.1 mM tiron did not produce any further reductions (Figure 3C,D).

Increased ROS levels and consequent oxidative stress inhibit osteoblastic differentiation of MSCs. However, our consistent results suggest that tiron, an antioxidant that eliminates oxidative stress in biological systems, directly exerts negative effects on osteoblastic differentiation of hPDCs, regardless of H_2_O_2_ treatment, which triggers cellular oxidative stress.

### 2.4. Effect of Tiron on ROS Generation in Osteoblast-Differentiating hPDCs

Considering that mitochondrial energy metabolism can regulate stem cell functions, we next examined the effects of tiron on mitochondrial function and ROS generation during osteoblastic differentiation of hPDCs. Although there was no significant change in ROS generation by day 3 of culture, ROS production was clearly increased in osteoblast-differentiating hPDCs by day 7 of culture (Figure 4A). Osteoclastogenesis is a multi-step process that requires the secretion of two cytokines that are required for bone resorption, macrophage colony-stimulating factor (M-CSF) and receptor activator of nuclear factor-kappa B ligand (RANKL), both of which are secreted by neighboring stromal/osteoblast-like cells. The binding of RANKL to its receptor RANK, which is expressed on the surface of osteoclast precursor cells, triggers osteoclast precursors to differentiate into osteoclasts [44,45,46]. Although ROS are known to be produced during RANKL-induced osteoclastogenesis in the bone marrow, our results suggest that ROS production may also be enhanced during osteoblastic maturation of hPDCs. In addition, some studies have suggested that ROS regulation is crucial for maintaining both MSC potency and MSC differentiation potential [47].

In a flow cytometry analysis, three different antioxidants (tiron, N-acetyl cysteine [NAC], and Mito-TEMPO) each significantly reversed the increased ROS production in osteoblast-differentiating hPDCs. Treatment with tiron or Mito-TEMPO reduced ROS production in the cells in a concentration-dependent manner, whereas 0.1 mM NAC and 1 mM NAC each reduced the generation of ROS in the hPDCs to a similar extent (Figure 4B).

Mitochondria generate approximately 90% of cellular ROS, and mitochondrial malfunction is usually accompanied by increased ROS levels, which impair the activities of MSCs. Mitochondria thus play an important regulatory role in determining the differentiation capacity of MSCs. MSCs have relatively low intracellular ROS levels, and low-to-moderate intracellular ROS levels are essential for cellular proliferation and differentiation [48,49]. Considering these points, our results suggest that mitochondria-targeted antioxidant activity can exert negative effects on MSC osteoblastic differentiation by causing mitochondrial dysfunction [50,51].

### 2.5. Effect of Antioxidants on Mitochondrial Metabolism in Osteoblast-Differentiating hPDCs

We next examined the effects of antioxidants on mitochondrial metabolism in osteoblast-differentiating hPDCs. Cellular metabolism is intimately linked to mitochondrial function. The oxygen consumption rate (OCR) is a key indicator of mitochondrial respiration, and OCR measurement provides information about metabolic function in cultured cells. Mitochondria are key for energy metabolism within cells, as they produce most of the cellular ATP through oxidative phosphorylation. Therefore, an impaired rate of ATP production reflects mitochondrial dysfunction [52,53,54].

We found that the OCR and the rate of ATP production increased over time in hPDCs during osteoblastic differentiation (Figure 4D,E). Treatment with 0.1 mM or 1 mM tiron, 1 mM NAC, or 0.1 mM Mito-TEMPO significantly reduced the OCR in osteoblast-differentiating hPDCs. Similarly, treatment with 1 mM tiron, 1 mM NAC, or 0.1 mM Mito-TEMPO significantly reduced the rate of ATP production in the osteoblast-differentiating cells (Appendix A).

Mitochondrial dynamics regulate metabolic shifts between glycolysis and oxidative respiration. In undifferentiated MSCs, mitochondrial activities are maintained at a low level, but glycolytic activities are consistently maintained at a high level for a majority of glycolytic enzymes and lactate production [55]. When stem cells commit to differentiation, glycolytic metabolism shifts to mitochondrial oxidative phosphorylation to meet the increased cellular energy demand of differentiated cells. Oxidative phosphorylation is achieved by five multimeric enzyme complexes that are present within the inner mitochondrial membrane. Active mitochondrial respiration provides ATP to drive cellular functions and is also needed to maintain the differentiation ability of stem cells [56,57]. Our results suggest that mitochondrial biogenesis is induced during the osteoblastic differentiation of hPDCs, which is associated with increased mitochondrial activity, and antioxidant-induced impairment of mitochondrial function may have unfavorable effects on the osteoblastic differentiation of MSCs.

### 2.6. Effects of Tiron on Mitochondrial Proteins in Osteoblast-Differentiating hPDCs

We next assessed the expression of several mitochondrial proteins in osteoblast-differentiating hPDCs treated with tiron by Western blot to better understand the impairment of mitochondrial biogenesis by antioxidants. The levels of the mitochondrial proteins were clearly increased in hPDCs cultured in osteogenic-induction medium compared with those in DMEM (Figure 5A,B). This suggests that mitochondrial mass is gained and mitochondrial biogenesis is induced when hPDCs undergo osteoblastic differentiation.

Although tiron did not affect the levels of HSP60 and VDAC in the osteoblast-differentiating hPDCs, it significantly reduced the levels of most other mitochondrial proteins, particularly cytochrome c and SDHA, regardless of its concentration (Figure 5A,B). Cytochrome c is a protein that functions as an electron carrier in the mitochondrial respiratory chain. SDHA encodes a major catalytic subunit of succinate-ubiquinone oxidoreductase, a complex of the mitochondrial respiratory chain. HSP60 plays pivotal roles in the regulation of protein folding and the prevention of protein aggregation. VDAC, located in the mitochondrial outer membrane, functions as a gatekeeper for the entry and exit of mitochondrial metabolites and has been used as an indicator of mitochondrial mass [58,59].

Further studies are needed to determine why mitochondrial HSP60 and VDAC are not affected by the antioxidant tiron; however, our results suggest that tiron directly affects the mitochondrial function of osteoblast-differentiating hPDCs and may indirectly inactivate other cellular functions, including osteoblastic differentiation, by causing mitochondrial dysfunction.

## 3. Materials and Methods

### 3.1. Culture and Differentiation of hPDCs

Periosteal explants (5 mm × 20 mm) were harvested from mandibles during surgical extraction of impacted lower third molars, and hPDCs were isolated as previously described [59,60]. Briefly, periosteal pieces were cultured at 37 °C in 95% humidified air and 5% CO_2_ in 100-mm culture dishes containing Dulbecco’s modified Eagle’s medium (DMEM) supplemented with 10% heat-inactivated fetal bovine serum (FBS), 100 IU/mL penicillin, and 100 μg/mL streptomycin. Osteoblast differentiation was induced by culturing periosteal cells (seeded at a density of 3 × 10^4^ cells/well on 24-well plates) at passage 3–5 in osteogenic induction medium composed of DMEM supplemented with 10% FBS, 50 µg/mL L-ascorbic acid 2-phosphate, 10 nM dexamethasone, and 10 mM β-glycerophosphate. The media were changed every 3 days. In the case of hPDC, 1 passage means the number of times the cells were detached by trypsinizing when the cells were 100% confluence after seeding 5 × 10^5^ in a 150-mm culture plate.

### 3.2. Measurement of hPDC Viability in the Presence of Tiron

To investigate the effects of tiron on the viability of hPDCs, we seeded hPDCs at a density of 3 × 10^4^ cells/well on 24-well plates in DMEM and then treated the cells with 0.1 mM, 1 mM, 10 mM, or 100 mM tiron (Sigma-Aldrich, St. Louis, MO, USA). The viability of the hPDCs was assayed using a (CCK)-8 cell counting kit (Dojindo Molecular Technologies, Rockville, MD, USA) and a previously published method [60,61,62].

### 3.3. Measurement of Mitochondrial Membrane Potential in hPDCs Treated with Tiron

To assess the changes in the mitochondrial function of hPDCs treated with tiron, the MMP was measured in cells at passages 5, 10, and 15 using a JC-1 assay kit (Abcam, Cambridge, MA, USA) according to the manufacturer’s instructions. The hPDCs were seeded on a 96-well black plate and incubated with tiron for 24 h. The osteoblast differentiation group was cultured in osteogenic induction medium for up to 7 days. Then, the culture medium was replaced by JC-1 solution and incubated at 37 °C for 10 min in the dark. Fluorescence intensities were then measured using a fluorescence microplate reader.

### 3.4. Senescence-Associated β-Galactosidase Activity and Staining

To determine whether tiron has a protective function against H_2_O_2_-induced oxidative stress, senescence in hPDCs was evaluated using a senescence β-galactosidase staining kit (Cell Signaling Technology, Danvers, MA, USA). Briefly, hPDCs were seeded on a 12-well plate and pretreated with tiron (1 mM) in DMEM for 24 h before stimulation with H_2_O_2_. Then, the cells were treated with 0.1 mM H_2_O_2_ and incubated overnight. The cells were then washed twice with PBS, and 1 mL fixative solution was added. After incubation for 15 min, the cells were incubated with 1 mL β-galactosidase staining solution in a dry incubator without CO_2_. On the following day, β-galactosidase-positive (blue-stained) cells were detected under a microscope.

### 3.5. Evaluation of Osteoblastic Differentiation of hPDCs Treated with Tiron

To investigate the effects of tiron on the osteoblastic differentiation of hPDCs undergoing senescence, hPDCs at passages 5, 15, and 25 in osteogenic induction medium were left untreated (control) or treated with 0.1 mM or 1 mM tiron. The osteogenic phenotypes of the hPDCs were then assessed as previously described [40]. Additionally, the osteoblastic cells were decalcified with 0.6 N HCl for 24 h at room temperature for the calcium deposition assay. The calcium content of supernatants was determined by spectrophotometry using a calcium assay kit (Abcam, Cambridge, MA, USA). Cells were stained with NBT/BCIP substrate solution (Pierce Chemical Co., Rockford, IL, USA) or 2% alizarin red S solution for histochemical detection of ALP and alizarin red S, respectively. The ALP activity was determined using TRACP & ALP assay kit (Takara Bio Inc., Shiga, Japan) according to the manufacturer’s instructions. ALP staining and activity assays were performed on day 10 of culture, whereas the determination of alizarin red S staining and calcium measurement were performed on day 20 of culture. Media were changed every 3 days, and tiron was added at each change of the medium.

### 3.6. Effects of Tiron on the Osteoblastic Phenotypes of hPDCs Treated with H_2_O_2_

To examine the effects of tiron on the osteoblastic phenotypes of hPDCs under oxidative stress, hPDCs in osteogenic induction medium were pretreated with 0.1 mM H_2_O_2_ for 12 h. Then, the cells were left untreated (control) or treated with 0.1 mM or 1.0 mM tiron and cultured for 20 days. We performed ALP staining and activity assays after 10 days of culture. We performed alizarin red S staining for mineralization and measured the calcium contents of the cells on day 20 of culture. Media were changed every 3 days, and tiron and H_2_O_2_ were added at each change of the medium.

### 3.7. Measurement of ROS in Osteoblast-Differentiating hPDCs

To examine the effects of tiron on mitochondrial function in hPDCs, we first observed the levels of ROS generated in osteoblast-differentiating hPDCs during 1 week of culture. To measure the cellular ROS levels, 3 × 10^4^ hPDCs were stained with 10 µM 2′,7′-dichlorodihydrofluorescein diacetate acetyl ester (DCFDA; Abcam, Cambridge, MA, USA) in 96-well black plate at 37 °C for 45 min, and fluorescence was measured (485 nm excitation, 535 nm emission) on a microplate reader after removal of DCFDA and addition of washing buffer according to the manufacturer’s instructions.

After confirming that the time when the ROS level started to increase statistically significantly was the first week of differentiation, osteoblast differentiation was induced with tiron for one week. All of the cells were then subjected to flow cytometry. For positive control, other antioxidants (NAC and Mito-TEMPO) were also used. First, 1 × 10^6^ hPDCs were grown on a six-well culture plate in the presence of 0.1 mM or 1 mM tiron, 0.1 mM or 1 mM NAC, or 0.01 mM or 0.1 mM Mito-TEMPO for 1 week in osteogenic induction medium at 37 °C under 5% CO_2_. The cells were then harvested to prepare a single-cell suspension, incubated with 20 µM DCFDA for 30 min at 37 °C, and washed with PBS. Flow cytometry was then performed with a BD LSRFortessa™ flow cytometer (Becton Dickinson, Franklin Lakes, NJ, USA), and data were analyzed using BD FACSDiva software.

### 3.8. Measurement of Metabolic Flux and ATP Production in Osteoblast-Differentiating hPDCs

To investigate the effects of antioxidants on mitochondrial metabolism in osteoblast-differentiating hPDCs, OCR and ATP production were analyzed in antioxidant-treated hPDCs. First, 5 × 10^3^ hPDCs were cultured on Seahorse XF-96 plates with osteogenic-induction medium in the presence of 0.1 mM or 1 mM tiron, 0.1 mM or 1 mM NAC, or 0.01 mM or 0.1 mM Mito-TEMPO for 1 week. The day before the Seahorse assay, sensor cartridges were hydrated in Seahorse XF Calibrant solution (Agilent Technologies, Santa Clara, CA, USA) overnight at 37 °C in a non-CO_2_ incubator. On the day of the assay (day 7 of culture), the cells were washed three times and incubated in XF Seahorse Base Medium DMEM supplemented with 25 mM glucose, 1 mM sodium pyruvate, and 4 mM L-glutamine to mimic cell growth and reperfusion conditions. The XF Seahorse base medium DMEM was prepared following the manufacturer’s instructions (Agilent).

OCR was calculated from Seahorse XFp Cell Mito Stress Tests according to the manufacturer’s protocol. The OCR reading was taken over time under basal conditions after the addition of mitochondrial inhibitors (1 µM oligomycin, 1 µM FCCP, and 0.5 µM rotenone/antimycin). Cellular ATP production rates were quantified using the Agilent Seahorse XF Real-Time ATP Rate Assay kit (Agilent Technologies) with label-free technology. OCR and ATP production rates were determined and analyzed on Agilent’s Seahorse Bioscience XF96 Extracellular Flux Analyzer (Seahorse Bioscience, North Billerica, MA, USA) according to the manufacturer’s instructions.

### 3.9. Western Blot Analysis

To extract all proteins, hPDCs were placed in RIPA buffer (Cell Signaling Technology, Danvers, MA, USA) containing a protease and phosphatase inhibitor cocktail. After 20 min, the cell pellets were sonicated and centrifuged, and the supernatant was resolved by sodium dodecyl sulfate-polyacrylamide gel electrophoresis (SDS-PAGE) followed by electrophoretic transfer onto a polyvinylidene difluoride membrane (Millipore, Burlington, MA, USA). The membrane was then probed with primary antibodies (#8674, Cell Signaling Technology) against human cytochrome C oxidase subunit IV (COX IV), cytochrome C, heat shock protein 60 (HSP60), prohibitin 1 (PHB1), pyruvate dehydrogenase (PDH), succinate dehydrogenase complex flavoprotein subunit A (SDHA), ALS-linked mutant superoxide dismutase 1 (SOD1), voltage-dependent anion channel 1 (VDAC), and β-actin (#3700, Cell Signaling Technology). Specific antibody binding was detected by horseradish peroxidase-conjugated secondary antibodies and then visualized using an enhanced chemiluminescence detection reagent (Invitrogen, Carlsbad, CA, USA).

### 3.10. Statistical Analysis

Each experiment was carried out at least three times independently. The results of one of three independent experiments are shown as representative data. Data are expressed as the mean ± standard deviation. Statistical analyses were conducted using GraphPad Prism 7.0 software (GraphPad Software, Irvine, CA, USA). We used one-way analysis of variance (ANOVA) with Tukey’s multiple comparisons (SPSS Statistics 22 software, IBM, Armonk, NY, USA) for all statistical analyses. We considered comparisons with *p* < 0.05 to be statistically significant.

## 4. Conclusions

As an antioxidant, tiron can alleviate cellular senescence by increasing the MMP and protecting against oxidative stress. However, tiron adversely affects senescence and osteogenic phenotypes during hPDC osteoblastic differentiation. In addition, tiron exerts negative effects on the osteoblastic differentiation of hPDCs even under H_2_O_2_-induced oxidative stress. Mitochondria play an important regulatory role in determining the differentiation capacity of MSCs. Tiron reduced the mitochondrial biogenesis induced by osteoblastic differentiation in hPDCs. Therefore, although antioxidants can improve MSC differentiation by reducing ROS generation, tiron caused mitochondrial dysfunction that had adverse effects on the osteogenic phenotypes of hPDCs. From the clinical point of view, we think that our study could have significance in preventing the use of antioxidants without sufficient screening regarding mitochondrial effects, especially in the elderly with relatively weak bone formation.

## Figures and Tables

**Figure 1 ijms-23-14040-f001:**
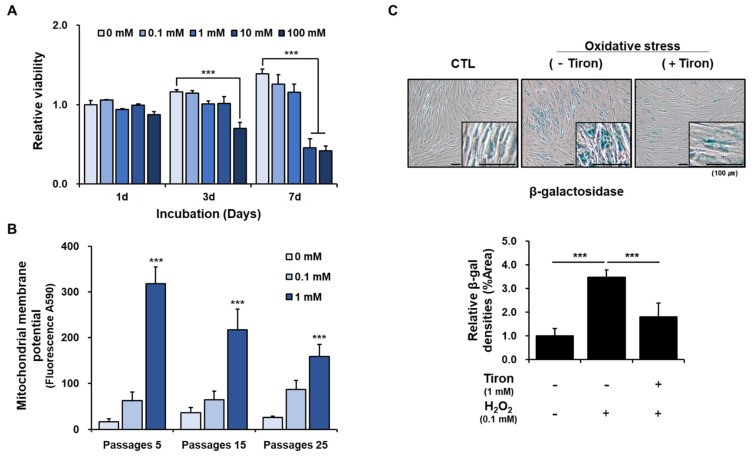
Biological activity of tiron in hPDCs. (**A**) The viability of hPDCs was unchanged after culture with 1 mM tiron compared to that of control cells. (**B**) Treatment with 1 mM tiron significantly increased the MMP in hPDCs, regardless of the passage. (**C**) Treatment with 1 mM tiron clearly decreased H_2_O_2_-induced β-galactosidase activity in hPDCs. (*** *p* < 0.001).

**Figure 2 ijms-23-14040-f002:**
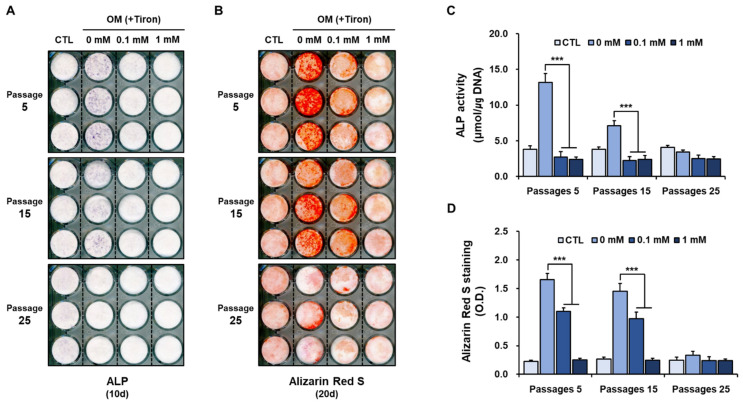
Effects of tiron on osteogenic phenotypes of hPDCs. Tiron reduced osteoblastic differentiation of hPDCs by reducing (**A**,**C**) ALP activity and (**B**,**D**) alizarin red-positive mineralization. CTL: DMEM; OM: osteogenic induction media. (*** *p* < 0.001).

**Figure 3 ijms-23-14040-f003:**
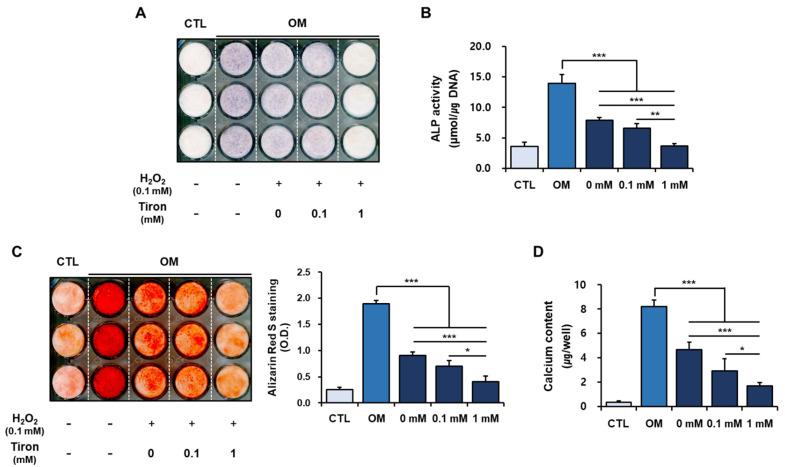
Effects of tiron on osteogenic phenotypes of hPDCs treated with H_2_O_2_. (**A**,**B**) Treatment with 1 mM tiron reduced ALP expression and activity in H_2_O_2_-treated hPDCs. (**C**,**D**) Alizarin red-positive mineralization and calcium content were clearly reduced in H_2_O_2_-treated hPDCs. CTL; DMEM., OM; osteogenic induction media. (* *p* < 0.05, ** *p* < 0.01, *** *p* < 0.001).

**Figure 4 ijms-23-14040-f004:**
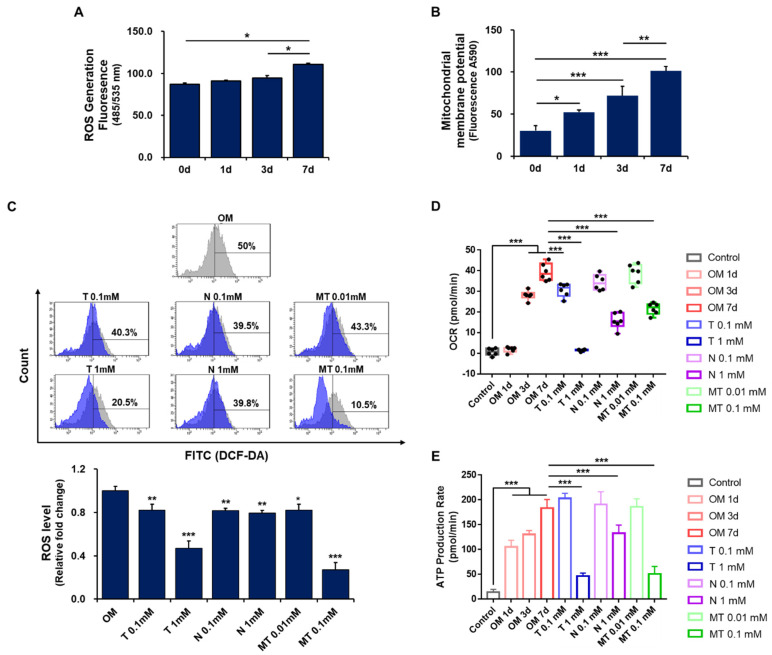
Effects of tiron on ROS production in osteoblast-differentiating hPDCs. (**A**) ROS production was significantly increased in hPDCs after 7 days of culture in osteogenic induction medium. (**B**) MMP was significantly increased in hPDCs in a time-dependent manner in osteogenic induction medium. (**C**) In flow cytometry analysis, antioxidants reduced ROS levels in hPDCs. The 50% value (percentage) was the control. (**D**,**E**) Effects of antioxidants on mitochondrial activity in osteoblast-differentiating hPDCs. Antioxidants significantly reduced the oxygen consumption rate (OCR) (**D**) and the rate of ATP production (**E**), which were both clearly increased during osteoblastic differentiation. Exceptions were treatment with 0.1 mM NAC that did not have a significant effect on OCR and treatments with 0.1 mM tiron, 0.1 mM NAC, and 0.01 mM Mito-TEMPO that did not have significant effects on the rate of ATP production. Control: DMEM; OM: osteogenic-induction medium; T: tiron; N: NAC; MT: Mito-TEMPO. (* *p* < 0.05, ** *p* < 0.01, *** *p* < 0.001).

**Figure 5 ijms-23-14040-f005:**
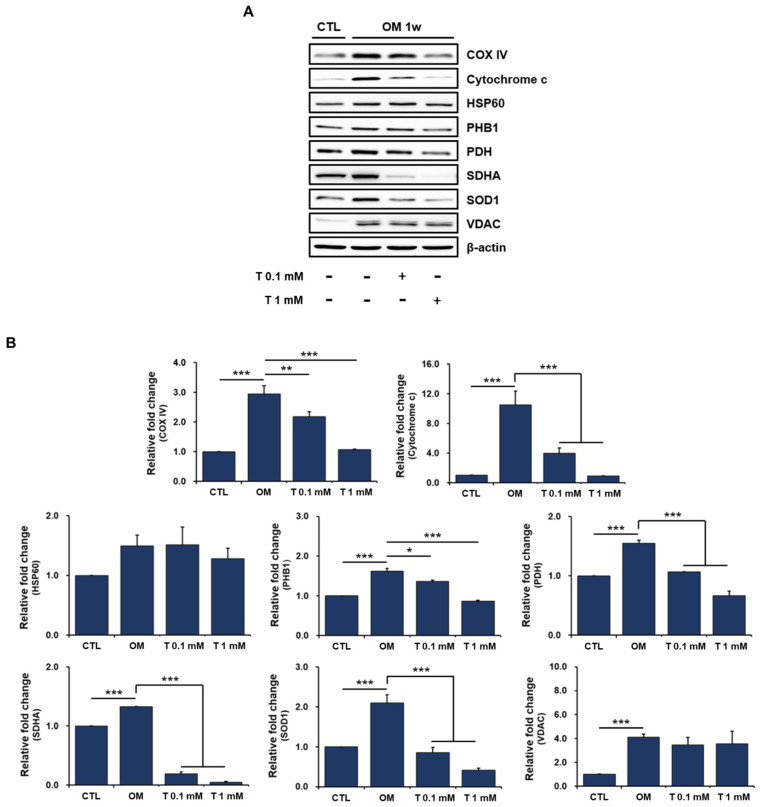
Effects of tiron on the level of mitochondrial proteins in osteoblast-differentiating hPDCs. (**A**,**B**) The levels of mitochondrial proteins were clearly increased in hPDCs cultured in osteogenic-induction medium compared with those in DMEM. Tiron markedly reversed the increases in mitochondrial protein levels induced by osteogenic induction medium, except for HSP 60 and VDAC. CTL: DMEM; OM: osteogenic induction medium; T: tiron. (* *p* < 0.05, ** *p* < 0.01, *** *p* < 0.001).

## Data Availability

The data that support the findings of this study are available from the corresponding author, upon reasonable request.

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
