# Peer review of "Tiron Has Negative Effects on Osteogenic Differentiation via Mitochondrial Dysfunction in Human Periosteum-Derived Cells"

_ijms, 2022, doi:10.3390/ijms232214040_

Round 1

Reviewer 1 Report

Thank you for describing the research study in an easier manner. 

Please enlarge figure 1C to clearly show the cells and all details in them, and put a bar to show the cell size.

Enlarge the figure 2A-B, and if possible, include their large images too to provide more details. From these images, it is very difficult to see whether there are enough and equal cells in each well or not. Cells could be washed away before or during the staining procedure. Please make sure.

hPDCs at passages 5, 15, and 25. Please clearly mention how the passages were counted.

One recently published research has shown that RAW264.7 is undergoing senescence due to serial passaging in vitro (Int. J. Mol. Sci. 202223(4), 2371; https://doi.org/10.3390/ijms23042371). Here, a similar effect is shown for hPDCs in this current manuscript. I think the authors can discuss this paper along with their findings in order to focus on the bone remodeling process (both bone resorption and formation). Another paper also focused on the effect of cell passaging on phenotypic and functional stability of RAW 264.7 cell (PLoS ONE 201813, e0198943.). 

Please include a graphical abstract if possible to clearly present the research findings.

Author Response

List of Responses to the Comments of Reviewer

We appreciate the detailed and valuable comments of the reviewer. We revised the followings as commented by the reviewer.

English language and style are fine/minor spell check required.

We had professional English proofreading before submitting our manuscript to Life from Bioscience Writers in Houston, USA. In addition, we also carefully checked the english language and style before submitting our revised manuscript

Reviewer 1

Comments and suggestions for authours

  1. Please enlarge figure 1C to clearly show the cells and all details in them, and put a bar to show the cell size.

: This part has been revised according to your comment.

  1. Enlarge the figure 2A-B, and if possible, include their large images too to provide more details. From these images, it is very difficult to see whether there are enough and equal cells in each well or not. Cells could be washed away before or during the staining procedure. Please make sure.

: We respect your opinion. However, this staining experiment is based on the provision of culture plate wells of all experimental groups for the overall and accurate evaluation.

In previous studies, we have often received feedback that partial cell staining images are unreliable. After that, to provide a clean background after staining, each well of the plate is blocked with cotton. Therefore, it is not possible to provide a magnified microscope image.

In addition, since these experiments are basically involved with the cell culturing for 10 to 20 days and then staining, cell confluence is 100% without exception. And, due to the characteristics of osteoblasts, they are deposited together with calcium at the bottom of the plate, so the probability that cells fall off the plates during the staining processes is very low. But from now on, according to your advice, before blocking with cotton, we will provide enlarged cell images with the whole images.

  1. hPDCs at passages 5, 15, and 25. Please clearly mention how the passages were counted.

: This part has been revised according to your comment.

  1. One recently published research has shown that RAW264.7 is undergoing senescence due to serial passaging in vitro (Int. J. Mol. Sci. 2022, 23(4), 2371; https://doi.org/10.3390/ijms23042371). Here, a similar effect is shown for hPDCs in this current manuscript. I think the authors can discuss this paper along with their findings in order to focus on the bone remodeling process (both bone resorption and formation). Another paper also focused on the effect of cell passaging on phenotypic and functional stability of RAW 264.7 cell (PLoS ONE 2018, 13, e0198943.).

: We identified two papers that you suggested. It is considered to be a paper on osteoclast differentiation, phenotypic and functional stability according to the passage of RAW 264.7 cells. In a previous study, we confirmed the decrease in osteoblast differentiation ability according to cell passage and focused on the aging-related bone formation process (figure). We will carefully review the follow-up studies with your proposed paper. Thanks for your suggestion.

  1. Please include a graphical abstract if possible to clearly present the research findings.

: We also want to include fancy creations of graphical abstracts. However, I hope you understand that we did not prepare the materials for the graphical abstract for this study. If it is not essential at this time, we will make sure to include it when submitting a follow-up study.

Respectfully yours,

June-Ho Byun

Reviewer 2 Report

Reviewers’ comments to authors

The manuscript entitled “Effects of tiron on osteogenic phenotypes and mitochondrial activity in human periosteum-derived cells” by Park et. al. describes the negative role of tiron in osteoblastic differentiation of human periosteum-derived cell (hPDCs), suggesting the association of mitochondrial dysfunction. However, there are limitations of this study, and the data are not sufficient to support their conclusion. My comments are as follow;

1. The title of this manuscript is not matching. It is not reliable to say the osteogenic phenotypes of hPDCs by examining only ALP and mineralization activity of this cells. Also, please mention a negative role of the agent in the changed title.

2. Please provide sufficient introduction about the tiron. For example, its broad application in bone-related diseases or any other diseases along with the intracellular signaling molecules specifically regulated/controlled by the tiron should be thoroughly and more detail described in the section of Introduction.

3. Authors examined that tiron had no direct protective effects on hPDCs’ senescence and osteoblastic differentiation. Considering the common knowledge on that antioxidants are generally capable of inhibiting cellular senescence, as well as enhancing in vitro differentiation of MSCs by reducing ROS generation, the results in this study are controversary. Please provide what are rationales of this negative result of the antioxidant. Also, the authors should discuss clearly what are the advantage of this work in the sense of clinical perspective.

4. Authors should show the results of mitochondrial membrane potential (MMP) during osteoblast differentiation to claim the results mentioned in the results section of 2.5 (last paragraph). Mitochondrial biogenesis is induced during the osteoblastic differentiation of hPDCs, which is associated with increased mitochondrial activity. However, results don’t fully verify that at the current state.

5. In Figure 1B, measuring only the fluorescence intensity of MMP don’t validate those results. They should show the fluorescence image stained by JC-1.

6. In Figure 4A, Authors should also present the ROS profile of this results as shown in figure 5B.

7. In Methods, Please describe briefly how the ALP and mineralization activity were examined. Also pls mention the catalog number of each antibody used for western blot.

8. Authors have isolated whole cell proteins in analyzing the mitochondrial-related proteins, so it was very hard to believe the results obtained from the western blotting. Why didn’t use the mitochondrial fraction of protein lysates.

9. In all figures, please provide a space between number and unit (ex, 1mM to 1 mM) like the text.

10. Pls represent the figure axis more exactly and adequately on the related figures or provide more detail explanation into the related figure legends. For example, in figure 1A, X-axis indicated 1 day 3 days 7 days, but what the day(s) means? Incubation day? If so, please describe the X axis with further reasonable way (ex, 1, 3, 7/Incubation (Days)) or explain that into the legends.

11. In Figure 1C, B-gal should be corrected to β-gal.

12. Pls provide further exactly what is the P5, P15, or P25 into the legends.

13. In the supplementary figure, pls provide further clearly the X and Y axis and also enlarge the font size in the Figure S1A.  

Author Response

List of Responses to the Comments of Reviewer

We appreciate the detailed and valuable comments of the reviewer. We revised the followings as commented by the reviewer.

English language and style are fine/minor spell check required

We had professional English proofreading before submitting our manuscript to Life from Bioscience Writers in Houston, USA. In addition, we also carefully checked the english language and style before submitting our revised manuscript

Reviewer 2

Comments and suggestions for authours

  1. The title of this manuscript is not matching. It is not reliable to say the osteogenic phenotypes of hPDCs by examining only ALP and mineralization activity of this cells. Also, please mention a negative role of the agent in the changed title.

: This part has been revised according to your comment.

  1. Please provide sufficient introduction about the tiron. For example, its broad application in bone-related diseases or any other diseases along with the intracellular signaling molecules specifically regulated/controlled by the tiron should be thoroughly and more detail described in the section of Introduction.

: According to your advice, the application in the diseases based on molecular mechanisms specifically regulated by tiron was further described in the introduction.

  1. Authors examined that tiron had no direct protective effects on hPDCs’ senescence and osteoblastic differentiation. Considering the common knowledge on that antioxidants are generally capable of inhibiting cellular senescence, as well as enhancing in vitro differentiation of MSCs by reducing ROS generation, the results in this study are controversary. Please provide what are rationales of this negative result of the antioxidant. Also, the authors should discuss clearly what are the advantage of this work in the sense of clinical perspective.

: Your remarks are very important. We also initially thought that tiron could enhance the in vitro differentiation of MSCs by reducing ROS production as an antioxidant. However, our results provided tiron has no direct effects on hPDC senescence and adversely affects hPDC osteoblastic differentiation. This is the main point of our experiments. Through further experiments, our study suggest that mitochondria-targeted antioxidant activity can exert negative effects on MSC osteoblastic differentiation by causing mitochondrial dysfunction.

This experiment is based on the in vitro cell level and suggests the opposite views about general cell aging and bone formation. Therefore, from the clinical point of view, we think that our study could have the significance in preventing the treatment of any antioxidants without sufficient screening about mitochondria, especially in the elderly with relatively weak bone formation. This part has been revised in the section of conclusions.

  1. Authors should show the results of mitochondrial membrane potential (MMP) during osteoblast differentiation to claim the results mentioned in the results section of 2.5 (last paragraph). Mitochondrial biogenesis is induced during the osteoblastic differentiation of hPDCs, which is associated with increased mitochondrial activity. However, results don’t fully verify that at the current state.

: An additional experiment (figure 4B) to measure MMP during osteoblast differentiation of hPDCs was performed and included in the manuscript.

  1. In Figure 1B, measuring only the fluorescence intensity of MMP don’t validate those results. They should show the fluorescence image stained by JC-1.

: Your comment makes a lot of sense. However, this data did not quantify the density of JC-1 fluorescence-stained cell images. The kit for this experiment is an assay through fluorescence spectrophotometer measurement according to the manufacturer's protocol. After seeding cells in 96-well black plate and staining JC-1, the overall fluorescence expression pattern of the cells was measured through a fluorescence microplate reader. We think that our study shows more specific and accurate results than showing partial fluorescence microscopy images of cells.

  1. In Figure 4A, Authors should also present the ROS profile of this results as shown in figure 5B.

: As described in the manuscript, figure 4A is a preceded experiment in which fluorescence measurement (485 nm excitation, 535 nm emission) was performed with a microplate reader according to the manufacturer's instructions to confirm the change in ROS production according to the osteoblast differentiation of hPDCs. After confirming the one-week period of differentiation in which ROS was significantly changed, the differentiation time was fixed at one week, and the amount of change in ROS due to antioxidant treatment was specifically confirmed through flow cytometry.

  1. In Methods, Please describe briefly how the ALP and mineralization activity were examined. Also pls mention the catalog number of each antibody used for western blot.

: This part has been revised according to your comment.

  1. Authors have isolated whole cell proteins in analyzing the mitochondrial-related proteins, so it was very hard to believe the results obtained from the western blotting. Why didn’t use the mitochondrial fraction of protein lysates.

: In general, we agree with you that the mitochondrial fraction should be used to analyze changes in mitochondrial-associated proteins within a short period of time after treatment of cells with drugs such as antioxidants. However, this study is always mediated by the condition of osteoblast differentiation of 7 days. It is known that when cells differentiate, mitochondrial metabolism basically changes or mass increases, and this fact has been verified in our research results.

Therefore, we performed western blotting by separating whole cell proteins in order to reduce the variables according to the mitochondrial fraction between the undifferentiated control group and the experimental group. In addition, this was confirmed in the whole cell protein because it focused on the overall mitochondrial change rather than selecting the effect of the mitochondrial specific complex (1-5) caused by tiron treatment.

In next studies, we will analyze the mitochondrial proteins using mitochondrial fractions through subsequent studies analyzing the specific mechanism of action of antioxidants.

  1. In all figures, please provide a space between number and unit (ex, 1mM to 1 mM) like the text.

: This part has been revised according to your comment.

  1. Pls represent the figure axis more exactly and adequately on the related figures or provide more detail explanation into the related figure legends. For example, in figure 1A, X-axis indicated 1 day 3 days 7 days, but what the day(s) means? Incubation day? If so, please describe the X axis with further reasonable way (ex, 1, 3, 7/Incubation (Days)) or explain that into the legends.

: This part has been revised according to your comment.

  1. In Figure 1C, B-gal should be corrected to β-gal.

: This part has been revised according to your comment.

  1. Pls provide further exactly what is the P5, P15, or P25 into the legends.

: This part has been revised according to your comment.

  1. In the supplementary figure, pls provide further clearly the X and Y axis and also enlarge the font size in the Figure S1A.

: This part has been revised according to your comment.

Respectfully yours,

June-Ho Byun

Round 2

Reviewer 2 Report

No more comments for authors.